# Rapid Progressive Fatal Acute Hemorrhagic Encephalomyelitis

**DOI:** 10.3390/diagnostics13152481

**Published:** 2023-07-26

**Authors:** Ssu-Yu Chen, Hung-Chieh Chen, Ting-Bin Chen

**Affiliations:** 1School of Medicine, Chung Shan Medical University, Taichung 40201, Taiwan; k95081@gmail.com; 2Division of Neuroradiology, Department of Radiology, Taichung Veterans General Hospital, Taichung 407219, Taiwan; 3School of Medicine, National Yang-Ming Chiao Tung University, Taipei 112304, Taiwan; sophiebeen@gmail.com; 4Department of Neurology, Neurological Institute, Taichung Veterans General Hospital, Taichung 407219, Taiwan; 5Dementia and Parkinson’s Disease Integrated Center, Taichung Veterans General Hospital, Taichung 407219, Taiwan; 6Center for Geriatrics and Gerontology, Taichung Veterans General Hospital, Taichung 407219, Taiwan

**Keywords:** acute hemorrhagic encephalomyelitis (AHEM), magnetic resonance imaging (MRI), acute disseminated encephalomyelitis (ADEM), microbleeds

## Abstract

Acute hemorrhagic encephalomyelitis (AHEM) is the most severe form of acute disseminated encephalomyelitis (ADEM). Patients with AHEM usually have unfavorable outcomes with high mortality rate. We reported a middle-aged male, who was diagnosed with AHEM and died 35 days after admission even under intensive immune therapy. Clinical courses were recorded and serial MR images were demonstrated to illustrate the rapidly changes in brain parenchyma. By highlighting these aspects, we hope to provide valuable insights for future studies and potential advancements in the management of AHEM.

A 56-year-old male was referred to our ER due to a progressive neurological dysfunction. Approximately two weeks before, the patient experienced episodes of blurred vision and headaches in both temporal regions, which occurred intermittently. A few days later, he suffered from the sudden onset of sensory aphasia, right facial palsy, and right hemiparesis, leading to the suspicion of a stroke, and was sent to another hospital. However, magnetic resonance imaging (MRI) revealed abnormal lesions at the bilateral parietal lobe, left frontal lobe, and right middle cerebellar peduncle, showing hyperintense on fluid-attenuated inversion recovery (FLAIR), hypointense on T1-weighted image (T1WI), faint contrast enhancement on Gadolinium-enhanced T1-weighted image (Gd-T1WI), no restricted diffusion on diffusion-weighted image (DWI), and the presence of microbleeds inside the lesion on suspected weighted image (SWI). Autoimmune disorders or demyelinating disease were considered (Figure 1). After three days of pulse therapy and the administration of oseltamivir, the patient’s condition improved. However, incoherent speech occurred two days later, and he was transferred to our hospital. Upon admission, the patient was still able to communicate and respond to questions. We repeated MRI at our hospital and MRI disclosed an enlargement of the right middle cerebellar peduncle lesion and new lesions at right frontal and right parietal lobes. The diffusion restriction of the lesions without obvious contrast enhancement was noticed this time. Additionally, the SWI sequence detected multiple microbleeds in the subcortical region of the left frontal lobe and right middle cerebellar peduncle (Figure 2). Pulse therapy was given. However, over time, the patient’s spontaneous verbal output diminished significantly, and he displayed a reduced responsiveness to sounds and pain. A subsequent MRI was performed nine days after the first MRI at our hospital due to conscious deterioration and poor response to previous treatment. The MRI showed a progressively increased number and size of the lesions and disseminated microbleeds (Figure 3). We then started plasma exchange and double filtration plasmapheresis. Even under intensive treatment, conscious change and low blood pressure occurred. An emergent CT scan was performed and showed diffused low-density changes in the brainstem and cerebrum, suggesting the presence of severe brain edema. These imaging findings indicated global hypoxia and a severely increased intracranial pressure (IICP) with brain herniation (Figure 4). In an effort to monitor the IICP, neurosurgeons implanted an ICP monitor, but unfortunately, the patient remained unresponsive even after the surgery. Despite the utilization of intensive surgical and immunological approaches, the patient passed away thirty-five days after admission.

AHEM commonly occurs in young adults and shows male predilection. There are several diseases that present with symptoms similar to AHEM, including myelin oligodendrocyte glycoprotein (MOG) antibody-associated disorder, neuromyelitis optica spectrum disorder (NMOSD), multiple sclerosis, infectious meningoencephalitis, vasculitis, progressive multifocal leukoencephalopathy (PML), Behçet syndrome and lymphomatoid granulomatosis [1,2]. MRI is crucial and could provide hints for the diagnosis of AHEM. Tumerfective lesions involving mainly cerebral white matter associated with brain edema and the presence of cerebral microbleeds are the key characteristics [3,4,5]. However, distinguishing AHEM from other similar diseases can still be challenging due to the absence of established guidelines for defining diagnostic algorithms. Clinical presentation, MRI findings, CSF analysis, and tissue biopsy play important roles in reaching a definitive diagnosis [6]. The blood autoimmune autoantibodies, aquaporin-4, CSF oligoclonal band, and CSF autoantibodies are all negative in our patient. Furthermore, there is no evident vessel stenosis observed using MR angiography. Therefore, vasculitis or any other autoimmune disorder is unlikely. Lymphomatoid granulomatosis, commonly referred to as angiocentric lymphoma, should be included in the list of potential diagnoses. This condition typically affects multiple areas of the brain’s white matter, displaying linear or punctate enhancement along the perivascular space [7]. In our case, rapid clinical deterioration despite intensive treatment, along with an elevated total protein level of 1.821 g/L and an increased red blood cell count of 9/μL in the CSF study, are typical features of AHEM as described in the literature. Although a tissue biopsy was not performed due to the family’s decision, the combination of clinical presentations, CSF tests, and MR imaging strongly suggests a high likelihood of AHEM. Due to its rapid progression, the prognosis of AHEM remains poor [8,9]. We hope that this article can serve as a valuable resource in illustrating brain imaging patterns in AHEM patients and contribute to a better understanding of this condition.

Axial non-contrast CT scan showed a diffused brain swelling of cerebrum and brainstem and an effacement of cortical sulci and suprasella cistern. Severe IICP is considered.

## Figures and Tables

**Figure 1 diagnostics-13-02481-f001:**
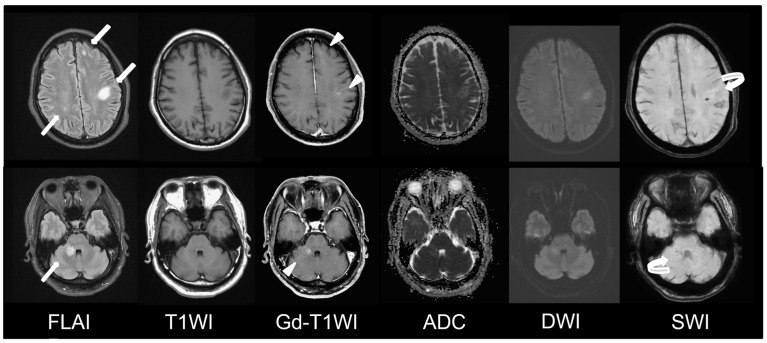
Initial MRI at other hospital. MRI fused images of axial FLAIR, T1WI, Gd-T1WI, ADC, DWI, and SWI images at supratentorial level (**top row**) and infratentorial level (**bottom row**). Abnormal hyperintense lesions are seen at left frontal lobe, left parietal lobe, right parietal lobe, and right middle cerebellar peduncle on FLAIR images (arrow). Contrast enhancement (arrowhead) and microbleeds (curved arrow) are seen in some of the lesions.

**Figure 2 diagnostics-13-02481-f002:**
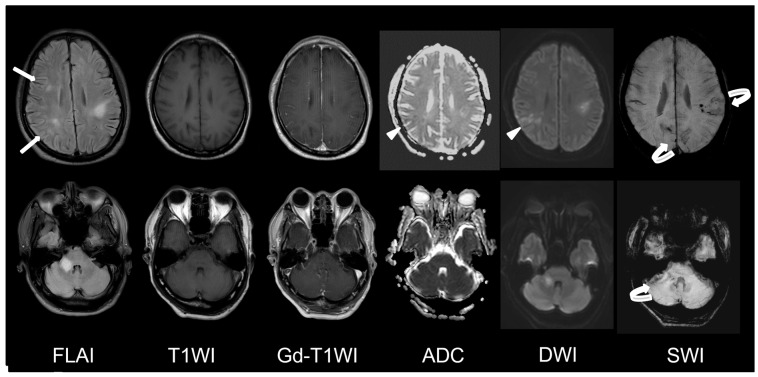
First MRI at our hospital. MRI disclosed an enlargement of right middle cerebellar peduncle lesion and new lesions at right frontal and right parietal lobes (arrow). Diffusion restriction of some lesions (arrowhead) and multiple microbleeds in the subcortical region (curved arrow) were seen.

**Figure 3 diagnostics-13-02481-f003:**
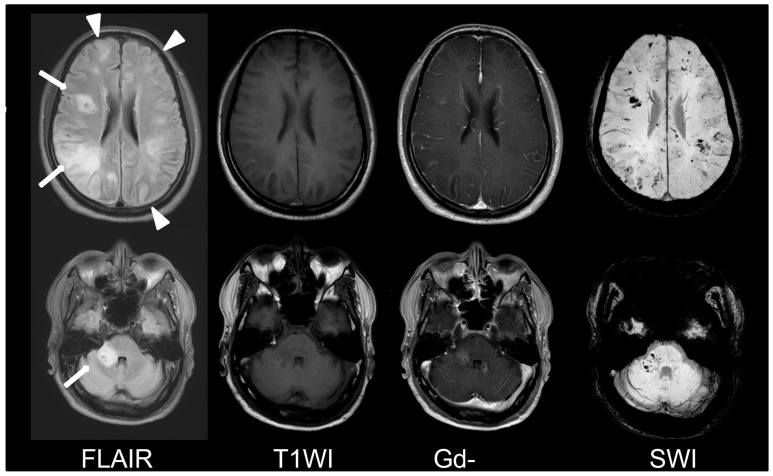
Second MRI at our hospital. Progressively increased number (arrowhead) and size (arrow) of the lesions and disseminated microbleeds were found.

**Figure 4 diagnostics-13-02481-f004:**
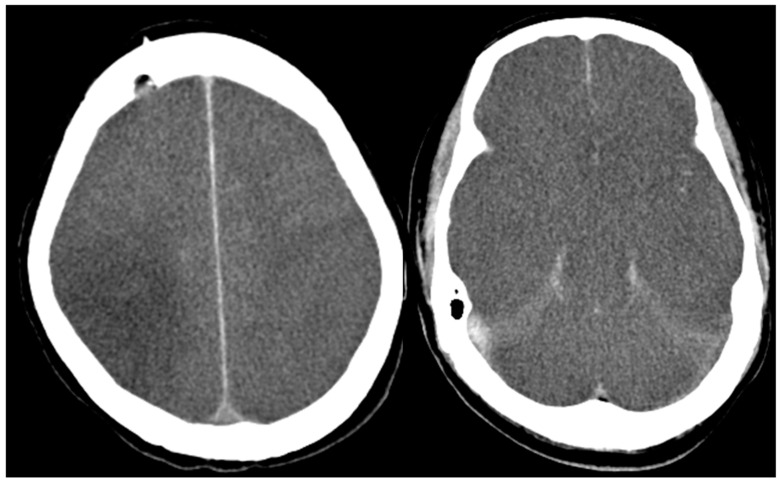
Non-contrast CT scan of brain. The images disclosed severe IICP with diffused brain swelling.

## Data Availability

No new data were created or analyzed in this study. Data sharing is not applicable to this article.

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
