# Peer review of "Rapid Progressive Fatal Acute Hemorrhagic Encephalomyelitis"

_diagnostics, 2023, doi:10.3390/diagnostics13152481_

Round 1
Reviewer 1 Report
The case is interesting and well illustrated, but some revision is required:
- in the description of Fig 1 (text 29-33) not all lesions are described, although one in the left frontal lobe even has contrast enhancement.
- it would be better to put the sequences names on the figures, as well as arrows to the main lesions.
- it s not clear how the final diagnosis was proofed - CSF analysis, post-mortem data et al? Othervise it could be difficult to say that it s not angiocentric lymphoma or vasculitis for example
Reviewer 2 Report
Review Report
· In this case report, the authors explored a rare case of middle age male, who was diagnosed with AHEM and died 35 days after admission.
· The case is interesting. The authors have worked hard to present this case report in a good manner. But I have one comment:
-Please write detailed description of the figures and share with arrows to the lesion.
